# Regulation of Tau Expression in Superior Cervical Ganglion (SCG) Neurons In Vivo and In Vitro

**DOI:** 10.3390/cells12020226

**Published:** 2023-01-05

**Authors:** Ying Jin, Theresa Connors, Julien Bouyer, Itzhak Fischer

**Affiliations:** Department of Neurobiology and Anatomy, Drexel University College of Medicine, Philadelphia, PA 19129, USA

**Keywords:** tau protein, Big tau, autonomic nervous system, neuronal development, microtubule-associated protein, exons, primary neurons, neuron injury, regeneration

## Abstract

The superior cervical ganglion (SCG) is part of the autonomic nervous system providing sympathetic innervation to the head and neck, and has been regularly used to prepare postnatal neuronal cultures for cell biological studies. We found that during development these neurons change tau expression from the low molecular weight (LMW) isoforms to Big tau, with the potential to affect functions associated with tau such as microtubule dynamic and axonal transport. Big tau contains the large 4a exon that transforms tau from LMW isoforms of 45–60 kDa to 110 kDa. We describe tau expression during postnatal development reporting that the transition from LMW tau to Big tau which started at late embryonic stages is completed by about 4–5 weeks postnatally. We confirmed the presence of Big tau in dissociated postnatal SCG neurons making them an ideal system to study the function of Big tau in neurons. We used SCG explants to examine the response of SCG neurons to lesion and found that Big tau expression returned gradually along the regrowing neurites suggesting that it does not drives regeneration, but facilitates the structure/function of mature SCG neurons. The structural/functional roles of Big tau remain unknown, but it is intriguing that neurons that express Big tau appear less vulnerable to tauopathies.

## 1. Introduction

Cultured SGC neurons have been regularly used as an in vitro model for sympathetic neurons and in studying axon growth and trafficking, synaptogenesis, microtubule dynamics, axon regeneration and degeneration and neuronal survival [1]. They can be grown as dissociated cells or explants and as a co-culture system with glial cells [2]. Sympathetic neurons can be prepared from embryonic or postnatal rats or mice using dissected SCGs and grown as pure neuronal cultures in the presence of nerve growth factor (NGF, [3,4]). These homogenous neuronal cultures can be processed for immunocytochemical staining, biochemical studies and physiological analysis [5,6]. SCG neurons extend axon-like neurites that rapidly grow to several millimeters making them an ideal system to study the properties of growing and degenerating axons growth in the context of the neuronal cytoskeleton [7]. What has not always been considered in these studies is the rapid changes in the expression of critical cytoskeletal protein, which can affect the properties of the neurons and interpretation of the results. In particular, there is a dramatic switch in the expression of tau from the low molecular weight (LMW) isoforms in early embryonic stages to Big tau at late postnatal stages which directly affects the dynamics of the microtubules as well as the properties of the axons with respect to growth, transport and possible degeneration [8,9].

The Big tau Isoform has been discovered 30 years ago [10,11,12] composed of the additional 4a exon which dramatically increased the size and likely the 3D structure of tau. Big tau was initially shown to be expressed in the peripheral nervous system (PNS) and later it has also been mapped to specific areas of the central nervous system that project to the periphery such as cranial nerve motor nuclei as well as retinal ganglion cells and the cerebellum [13]. There remains however a significant gap of knowledge about the developmental changes in tau expression with respect to the switch from the LMW isoforms to Big tau, the differential effects that it has on microtubule dynamics and the properties of axons that express this isoform. So far, it is known that although the 4a exon defining Big tau has a stable size of about 254-aa in a wide range on vertebrates, its sequence is not conserved among species; for example, showing almost no homology between mammals and non-mammals vertebrates [14,15].

The present study is designed to be a roadmap for the expression of the tau isoforms during development and in culture following the timeline of the switch to Big tau. This could serve as a reference for future studies taking into account the critical forms of tau expression particularly when studying microtubule dynamic and the properties of axons in growth, regeneration and degeneration. Using antibodies for all tau isoforms and those specific for Big tau we have presented structural, immunocytochemical and developmental data of SCG in vivo and in vitro including dissociated neuronal cultures and explants response to injury.

## 2. Experimental Procedures

### 2.1. Tissue Preparation

For morphological analysis, adult Sprague Dawley rats (4 months-old) were perfused transcardially by 0.9% normal saline followed by ice-cold 4% paraformaldehyde (PFA) fixative in phosphate buffer. All tissues needed for this study (SCG and DRGs, dorsal root ganglia) were dissected out and kept in same fixative overnight, and then transferred into 30% sucrose for 3–5 days. For SCG dissection, SCG was identified on the underside of a bifurcation of the carotid artery. Under dissecting microspore, the skin at the throat was cut from the chin to the upper chest. The fat pad around the sides of neck was cut away first. To visualize the carotid artery, three muscles needed to clip away by first removing the superior-most muscle (closer to head), then the second one (close to the low chest) and finally the muscle lying parallel to the trachea. Using forceps to hold the artery at the point of the bifurcation, SCG was underneath the bifurcation and could be completely removed. Tissues were embedded in M1 and cut either cross or sagittal sections at 20 µm. Sections were collected on Superfrost Plus (Fisher Scientific, Hampton, NH, USA) slides and kept at −20 °C.

For Polymerase chain reaction (PCR) and Western blots, fresh tissue was dissected from adult rat brain, DRGs, and SCGs at 1–7 weeks-old postnatally, adult (4 months-old), as well as aged (over 12 months-old), then transferred quickly into 1.5 mL tube with rNase, dNase Pyrogen-free (Capital Scientific, Inc., Austin, TX, USA) on dry ice. All samples were stored at −80 °C. For analysis of SCG explant, samples were collected following culturing for 8 days and 4 h or 5 days after lesion in the same 1.5 mL tubes and stored at −80 °C.

### 2.2. Primary SCG Culture

SCGs were dissected from rats at different ages: postnatal days 3, 14, 21, 28. To dissociate the tissue into single cells, SCGs were cut into small pieces and treated enzymatically with incubation in collagenase for 15 min at 37 °C followed by trypsin for 45 min also at 37 °C. Reaction was stopped by adding blocking medium (L-15 base medium, 1.2% D-glucose, 2% Glutamax, 2% Pen/strep, penicillin-streptomycin, 10% FBS, fetal bovine serum). SCGs were transferred into a 1.5 mL tube after removal of blocking medium, added 1 mL of plating medium (blocking medium with NGF 0.1 ng/mL), triturating the ganglia 10–15 times with P1000 pipette (Eppendorf North America, Enfiled, CT, USA) and then plated in 120 μL as suspension in a 24-well plate with coverslips coated with poly-D-lysine, (PDL, 1 mg/mL) and laminin (10 μg/mL) using 6–8 coverslips in a plating density of 4000 cells/coverslip. Cells were cultured for 3–10 days, and then fixed with 4% PFA for 30 min at room temperature. Coverslips were then washed 3 times with phosphate-buffered saline (PBS), and then stored at 4 °C for immunocytochemical staining.

### 2.3. SCG Explant Culture

SCGs were dissected from postnatal day 3 (P3) rats. Ganglia were cut into two pieces and placed on a 35 mm dish with 14 mm glass bottom pre-coated with PDL and laminin same as described above with limited culture medium for 2 h. More culture medium was added 2 h after incubation. Medium was changed every other day. After 8 days of culturing, neurites from SCG explant were cut using a 30-gauge needle and allowed to continue to grow. SCG explants were fixed with 4% PFA at different times after the lesion (4 h and 5 days) and washed with PBS 3 times. SCG explants in PBS were stored at 4 °C for staining with a Big tau specific antibody 1:1000 (Fischer lab, Philadelphia, PA, USA) described in Boyne et al., 1995 and a Tuj antibody (Covance, Biolegend, San Diego, CA, USA), 1:500).

To analyze the ratio of Big tau/Tuj staining in different regions of the explants, images were processed using Image J (NIH, Bethesda, MD, USA) as described in [16]. Briefly, we defined “proximal” areas inside 150 µm distance from the cell body and “distal” outside this area outlining 3 regions of 100 × 100 pixels for each area where the densities of Big tau and Tuj staining were measured and expressed as a ratio of Big Tau to Tuj. Values of the 3 regions of each image were averaged and Student’s *t*-test (SigmaPlot 13, Palo Alto, CA, USA) was performed to determine the significance of the differences between the proximal and the distal ratio in both time points (4 h and 5 days post lesion).

### 2.4. Western Blots

Tissue homogenates were disrupted and homogenized in cold lysis buffer (RIPA: 25 mM Tris-HCl pH 7.6, 150 mM NaCl, 1% NP-40 or Triton X-100, 1% sodium deoxycholate, 0.1% SDS) in the presence of protease inhibitor cocktail using a Fisher brand Sonic Dismembrator. Protein determination was done by Pierce™ BCA Protein Assay Kit (Thermo Fisher, Waltham, MA, USA). The tissue homogenates were stored at −80 °C. Equal amounts of protein from each sample were denatured in 4×protein sample loading buffer (LI-COR) containing 2-mercaptoethanol and boiled at 95 °C for 5 min. Samples and 2 μL of the Chameleon duo pre-stained protein ladder (LI-COR, Lincoln, NE, USA) were separated by sodium dodecyl sulfate polyacrylamide gel electrophoresis (SDS-PAGE) on 4%–12% polyacrylamide bis-tris gels (Invitrogen, Waltham, MA, USA). After distilled water washing and dry transfer (Thermo Fischer iBlot system, Waltham, MA, USA), PVDF membranes were washed in Tris-buffered saline pH 7.4 (TBS) and then blocked in Intercept TBS blocking buffer (LI-COR) for 1 h at room temperature, followed by incubation with primary antibodies: Big tau ([13] diluted 1:1000), 3′ Tau [9] diluted 1:5000, actin (Millipore, Burlington, MA, USA, 1:10,000) at 4 °C overnight. Membranes were washed with TBS and then incubated with fluorescently labeled IRDye secondary antibodies (LI-COR) for 1 h at room temperature. After several TBS washes, membranes were washed in distilled water and digital fluorescent visualization of signals were detected at 700 and/or 800 nm channels using the Odyssey^®^ CLx Imaging System (LI-COR). All primary and secondary antibodies were diluted in Intercept Antibody Diluent (TBS). To assess the levels of Big tau and LMW tau during development, the average densities of the corresponding bands from the original raw data of early (weeks 1–3) and late postnatal times (weeks 5–7) were calculated and plotted.

### 2.5. Polymerase Chain Reaction (PCR)

RNA extraction and purification was done using a RNeasy Mini Kit (Qiagen, Germantown, MD, USA). cDNA was obtained using High-Capacity cDNA Reverse Transcription kit (Applied Biosystems, ref. 4368814, Thermo Fisher Scientifics, Waltham, MA, USA). Primers were designed with the BLAST program to include sequences of exon 2 (5′—TCA GAA CCA GGG TCG GA—3′) to exon 4A (5′—GCA GGT TGC TTG TCA GTT GG—3′), exon 4A (5′—GGG TTC CAT CCC ACT TCC TG—3′) to exon 6 (5′—GGT GGT TCA CCT GAT CCT GG—3′); and exon 9 (5′—CCG TCT GCC AGT AAG CGC—3′) to exon 12 (5′ –CTA CCT GGC CAC CTC CTG GC—3′).

All primers were purchased from IDT (Integrated DNA Technologies, Inc., Coralville, IA, USA). PCR were run using a Taq DNA polymerases Kit (ABM, Richmond, BC, Canada) on a thermal cycler (Bio-Rad, Hercules, CA, USA).

### 2.6. Immunocytochemical/Histochemical Staining

Immunocytochemical tissue analysis was performed with the following antibodies used at single or double staining: Big tau 1:1000 (Fischer lab), 3′-Tau 1:1000 (Fischer lab), Tuj 1:500 (Covance), IB4 1:2000 (Invitrogen, Waltham, MA, USA), CGRP 1:2000 (Peninsula, Rheinstrasse, Switzerland), Parvalbumin 1:500 (Abcam, Waltham, MA, USA). Sections were washed 3 times in PBS and incubated with blocking buffer (PBS with 0.3% triton X-100, 0.5 mg/mL BSA, 0.01% thimerosal, 5–10% goat serum) for 1 h at room temperature, then incubated overnight with the primary antibodies for either single or double staining. On the following day, sections were washed 3 times in PBS, 10 min each, incubated with goat anti-mouse or goat anti-rabbit antibodies conjugated to FITC or rhodamine red (Jackson ImmunoResearch, West Gove, PA, USA) for 2 h, and then cover-slipped with anti-fade fluoromount-G (SouthernBiotech). All the staining experiments were repeated to assure the rigor of our data using a representative figure to illustrate the results.

For analysis of dishes or coverslips cultured with cells, cells were washed with PBS 3 times, incubated with blocking buffer for 15 min, and then incubated with the primary antibodies: Big tau 1:1000, Tuj 1:500 (Covance), and MAP1B 1:1000 (Fischer lab) for 1–2 h. Coverslips were washed with PBS, and then incubated with secondary antibodies: goat anti-mouse or goat anti-rabbit conjugated to FITC or rhodamine red (Jackson ImmunoResearch, West Grove, PA, USA) for 30 min, and then with anti-fade fluoromount-G (SouthernBiotech, Birmingham, AL, USA).

Slides or coverslips were visualized using a Leica DM5500B fluorescent microscope (Leica Microsystems, Wetzlar, Germany) with a Retiga-SRV camera (QImaging, Tucson, AZ, USA) and selected images were captured using Slidebook software (Olympus, Tokyo, Japan).

### 2.7. Statistical Analysis

SigmaPlot software package (Inpixon HQ, Palo Alto, CA, USA) was used to perform statistical analysis (one-way Anova, Student *t*-test) with a power of 0.05.

## 3. Results

### 3.1. Expression and Structure of Big Tau

The expression of Big tau in adult SCG tissue is illustrated by Western blot analysis in Figure 1A,B) using two sets of antibodies, the 3′tau that was prepared against the conserved C-terminal and recognizes all isoforms of tau (Figure 1A) and the other which was prepared against the complete 4a exon (Figure 1B) and recognizes only Big tau [9,13]. It is important to note that the Big tau antibodies were prepared against the recombinant 4a exon in bacterial expression vectors and thus had no cross reactivity with other isoforms of tau. The results with the 3′ tau antibodies show the expected LMW isoforms (45–60 kDa) detected in brain (Figure 1A), while only the Big tau isoform of 110 kDa is detected in SCG and DRG (Figure 1A,B). These results were verified with the Big tau-specific antibodies which show the expression of Big tau in the SCG and DRG samples. PCR analysis verified the expression of Big tau in SCG and DRG showing the presence of the 4a exon as well as exons 2/3 in the N-terminal and exon 10 defining a 4R structure with 4 microtubule binding domains (Figure 2A.1–A.3). While the presence of Big tau in SCG and DRG and LMW isoforms in brain is conclusive, we cannot exclude low level expression of the complementary tau isoforms. For example, our previous studies [13] have shown that Big tau is expressed in few selective areas of the central nervous system that project to the periphery such as cranial nerve motor nuclei, selective neurons in the cerebellum and retinal ganglion cells. Indeed, when enriching for these regions or overloading gels from brain (data not shown) there is a faint band of Big tau. Similarly, some peripheral neurons may continue to express low levels of LMW tau isoforms or as shown for DRG, not all neurons express Big tau and are therefore likely to express the LMW isoforms (Figure 5 and Appendix A).

### 3.2. Developmental Regulation of Tau Expression in SCG

Next, we analyzed the developmental regulation of tau expression in SCG neurons paying particular attention to the transition from the LMW to the Big tau isoforms. We found that in the first 2 weeks postnatally we could detect high levels of LMW tau at 45–60 kDa which were reduced at week 3 and remained at very low levels at week 5–7 (Figure 3A). Altogether, the average levels of the LMW isoforms at week 1–3 relative to week 5–7 were reduced by about 75% (Figure 3A1). At the same time the levels of Big tau were low at week 1 and increased 2-fold in subsequence weeks, so that by week 5–7 they were the dominant tau isoform (Figure 3A,A1). Interestingly, there is a period in early postnatal development where both LMW tau and Big tau are co-expressed defining a transition to the young adult state. In previous studies [13] we found that at embryonic age E14, SCG neurons expressed only the LMW isoforms of tau indicating that the transition to Big tau started at late embryonic stages and concluded by the first month postnatally. Big tau remained the dominant form in adult in our analysis of tissue from 4–12 months. Samples in the last two lanes in Figure 3A,B were collected from dissociated 5 weeks SCG cultured for 4 days (c-4d) and 7 days (c-7d). Both cultured SCG neurons expressed Big tau (Figure 3A,B) underscoring their potential value as a model system for studying Big tau regulation and function.

### 3.3. Histological Analysis of Big Tau

When SCG tissue is double stained with Big tau and Tuj antibodies (Figure 4) the results indicate that all the neurons express Big tau (Figure 4A,D). The Tuj antibody (Figure 4B,E) which detect neuronal beta III tubulin served as controls for the identification of the neurons and as a companion of tau—a microtubule associated protein (MAP). In contrast, staining of DRG neurons (Figure 5) showed selective expression with only small and medium size cells expressing Big tau (Figure 5A,D) as previously reported [13]. The selective expression is also apparent when the DRG are double stained for Big tau and markers of peptide neurotransmitters (CGRP, IB4) or parvalbumin (Appendix A).

### 3.4. Expression of Big Tau in Cultured Postnatal SCG Neurons

To investigate the expression of Big tau in cultured SCG neurons we used the standard protocol deriving the neurons from P3 [9] showing that after growing for 4 days the neurons formed long neurites and stained with the Big tau antibodies (Figure 6A) as well as MAP1B an axonal marker (Figure 6D). Note however, that at this stages SCG derived from the first postnatal week expresses both LMW tau and Big tau (Figure 3) and Figure 1 in [9]. SCG cultures derived from P28 and cultured for a week also expressed Big tau (Figure 7A,D) likely as the dominant isoform at that late developmental stage (Figure 3).

### 3.5. Analysis of Big Tau in SCG Explants

The next set of experiments required that we switch from dissociated SCG cultures to explants, where dissected tissue is grown in dishes and includes not only neurons but also support cells representing a more physiological model of growth and regeneration. Explant cultures also allowed us to collect enough tissue for Western blot analysis to demonstrate the expression and distribution of Big tau in these cultures (Figure 8D). Explants derived from P3 were grown for 8 days to develop an extensive network of neurites that grew around the explant (Figure 8A) and then lesioned at about half of their length (Figure 8B,C) and allowed to regrow (Figure 9 and Figure 10). Figure 9 shows the result of Tuj and Big tau expression 4 h after lesion (Figure 9B,C,F,G). The Tuj staining was observed in both proximal and distal regions of the neurites while Big tau staining was more prominent in the proximal region, best shown in (Figure 9D,H) of double staining. Five days later Big tau staining (Figure 10C,G) was similar to Tuj (Figure 10B,F) present in both proximal and distal regions of the neurites. Analysis of the ratio of Big tau/Tuj 4 h after lesion (Figure 9I–K) showed significantly lower ratios in distal relative to proximal regions of the neurites (*p* = 0.02 *t* TEST). In contrast, although the ratios of Big tau/Tuj 5 days after the lesion (Figure 10I–K) were still lower in distal relative to proximal regions of neurites, the differences were not significant (*p* = 0.22 *t* TEST).

## 4. Discussion

The studies presented here analyzed the expression of Big tau in SCG tissue and cultures setting a time frame for the transition of the tau isoforms during development and accordingly the properties of the neuronal cultures derived from different stages of development. The transition from LMW tau to Big tau starts during embryonic stages and is completed at about week 4–5 postnatally setting the SCG axons with an isoform of tau characteristic of PNS neurons (e.g., DRG). It also defined a postnatal stage where both LMW tau and Big tau a co-expressed. Why is this dramatic transition necessary remains a matter of speculation [8], but what is clear and can be informative is that the 4a exon defining Big tau is very large with little homology across the different vertebrate species. Indeed, our recent study [15] which analyzed the exon structure of MAPT (microtubule-associated protein tau**,** the tau gene) in vertebrates revealed that Big tau containing the 4a exon was present early in vertebrate evolution with low sequence conservation despite a stable size range of about 250aa. It suggested that the appearance of the significantly larger isoform of tau may have repeated during evolution, underscoring the importance of Big tau in supporting novel physiological functions required for a complex nervous system. For example, it is possible that the expression of Big tau is an adaptation of neurons with long axons that project to the periphery (e.g., PNS) to achieve robust and efficient axonal transport by acting as a much larger spacer between neighboring microtubules (the projection domain is increased from about 200 aa to >500 aa). Accordingly, the cross-sections of axonal microtubules have shown a spacing of ~35 nm when induced by Big tau compared to ~20 nm when induced by LMW tau [17]. Another provocative idea is that that 4a exon of Big tau changes the folding of the protein in a way that avoids the generation of aggregates and thus provides a protective mechanism to otherwise vulnerable neurons This idea is consistent with the metabolic stress of long projecting axons and the reduced or late appearance of tauopathy symptoms related to aggregation observed in these neurons. Indeed, one can also point out to areas of the CNS such as the cerebellum, which express Big tau and appear to be less vulnerable to tauopathies. The PCR analysis of the Big tau transcript of SCG have not only confirmed the presence of the 4a exon but have also revealed the presence of exons 2/3 and 10. The latter, which is one of the microtubule binding domains (MTBD) indicates that Big tau has the 4R isoform, which is typical of the adult tau.

Besides the general characterization of tau expression in SCG we expect that this work will also align the results and interpretation of cell biological studies using SCG cultures with the expression of the specific isoforms of tau. For example, a standard protocol of SCG cultures is using neurons derived from early postnatal time points when they express both LMW tau and Big tau [9,18,19,20,21,22,23,24]. In these and many other studies few have considered the unusual properties of SCG cultures where microtubules are associated with the unique isoform of Big tau even when studying axonal properties closely related to cytoskeleton structure and function. In practice, antibodies specific to Big tau have not been commercially available and the story of Big tau that originated in the 90′s has faded away. In the few cases that Big tau antibodies were available or when the analysis of tau expression included biochemical or molecular methods using Western blots and PCR, respectively [9] the presence of Big tau was noted. The challenge then and now remains how to compare and interpret the results of neurons expressing different tau isoforms. There has certainly been impressive progress studying the 6 variants of the LMW tau particularly when dealing with the alternative splicing of exon 10, which is one of the MTBD, as well as the relationship between the expression of tau variants and tauopathies [25,26,27,28,29]. Big tau however remains in the twilight of speculation about its physiological function and relationship to tauopathies. 

In contrast to the uniform staining of SCG, representing autonomic system neurons, DRG neurons, representing sensory neurons, express Big tau selectively reflecting the known heterogeneity of the sensory neurons and their diverse function [30]. Indeed, recent molecular analysis of DRG neurons [31,32] expanded the unique profile of many subtypes. It is therefore interesting that SCG neuron which provide sympathetic innervation to different areas of the head and neck (e.g., pineal gland, the choroid plexus, the eye, carotid body and the salivary and thyroid glands) and may have different phenotypes [6] all express Big tau. Both sensory and autonomic system neuron originate from the neural crest, both start expressing Big tau after E16 [13] but they differentiate and specialize in distinct pathways. We believe that elucidating the function of Big will shed light on the similarity and difference among these neurons in the context of the neuronal cytoskeleton in general and tau function in particular.

In the last part of this study, we used SCG explants to analyze the expression of Big tau during growth and regrowth following lesion. Although our lesion/regeneration experiments with the explants shown in Figure 8, Figure 9 and Figure 10 are preliminary and will need confirmation in both in vitro and in vivo systems, the results show that the explants are capable of rapid and robust regrowth of their neurites following a lesion. While the regrowing neurites contain microtubule, Big tau expression seems to be lagging and does not show at the distal neurite at 4 h but is there at the later timepoint 5 days after the lesion. These results are consistent with previous studies showing the expression of Big tau only in mature peripheral neurons (sensory and autonomic) and that the sequestration of Big tau in SCG cultures does not impair their growth [9]. Indeed, in regeneration experiments of sciatic nerve crushing the levels of Big tau have been reduced during the regeneration phase [33]. What then drive the growth/regeneration of the explants shown in Figure 9 and Figure 10? Most likely MAP1B, which is expressed in the SCG cultures (Figure 6), and which has been shown to be associated with axon growth in both the CNS and PNS neurons [34]. It seems therefore that mature SCG and other neurons that express Big tau retain endogenous growth associated proteins as well as an environment of Schwann cell conducive for regeneration. In contrast CNS neurons have a reduced capacity for regeneration but a better capacity for plasticity associated with the complex structure and function of the brain. The story of the selective expression of Big tau can therefore be an instructive example of the tradeoff in the balancing works of evolution.

## Figures and Tables

**Figure 1 cells-12-00226-f001:**
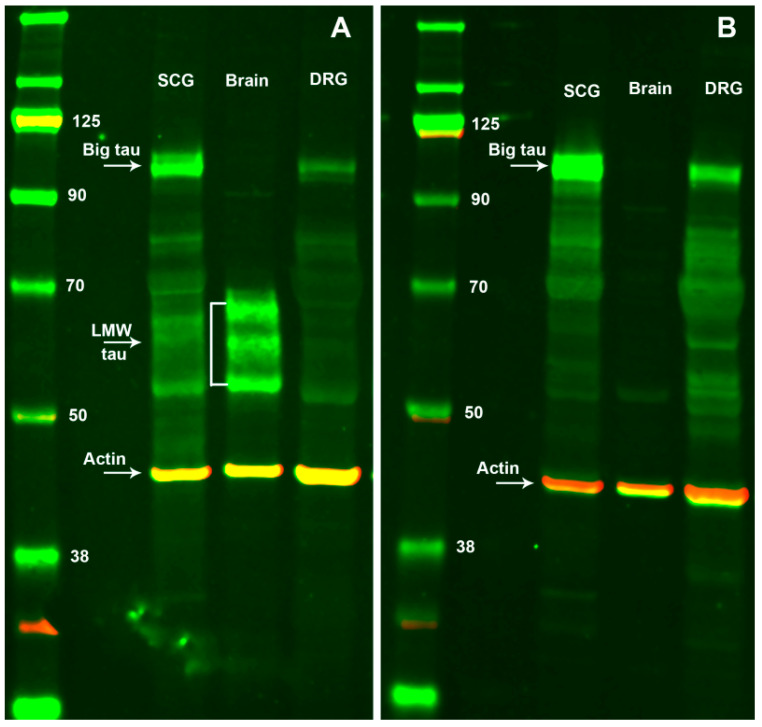
Expression of tau isoforms in SCG compared to brain and DRG tissues. Western blots stained with 3′tau antibodies (recognize all tau isoforms) show in (**A**) the low molecular weight isoforms of tau (LMW, 45–60 kDa) in brain only, while Big tau (110 kDa) is detected in SCG and DRG but not in brain. Western blots stained with the Big tau specific antibody, shown in (**B**), verified Big tau expression in SCG and DRG and its absence in brain. The blots included a pre-stained protein ladder and were probed for expression of actin (42 kDa) as a marker for loading variations.

**Figure 2 cells-12-00226-f002:**
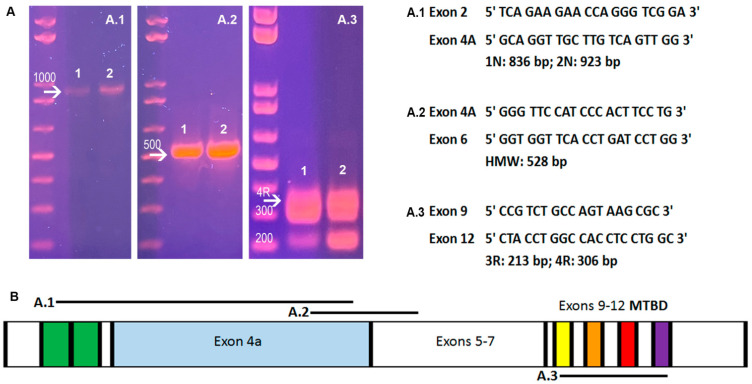
Exon structure of Big tau in SCG and DRG. (**A**) shows PCR analysis of DRG (1) and SCG (2) indicating the presence of exons 2 and 3 at the N-terminal (**A.1**), exons 4A and 6 (**A.2**); and exon 10 indicating a 4R structure (**A.3**). The sequence of the primers shown on the right side of the PCR images. (**B**) shows a graphical representation of Big tau exons and the PCR products.

**Figure 3 cells-12-00226-f003:**
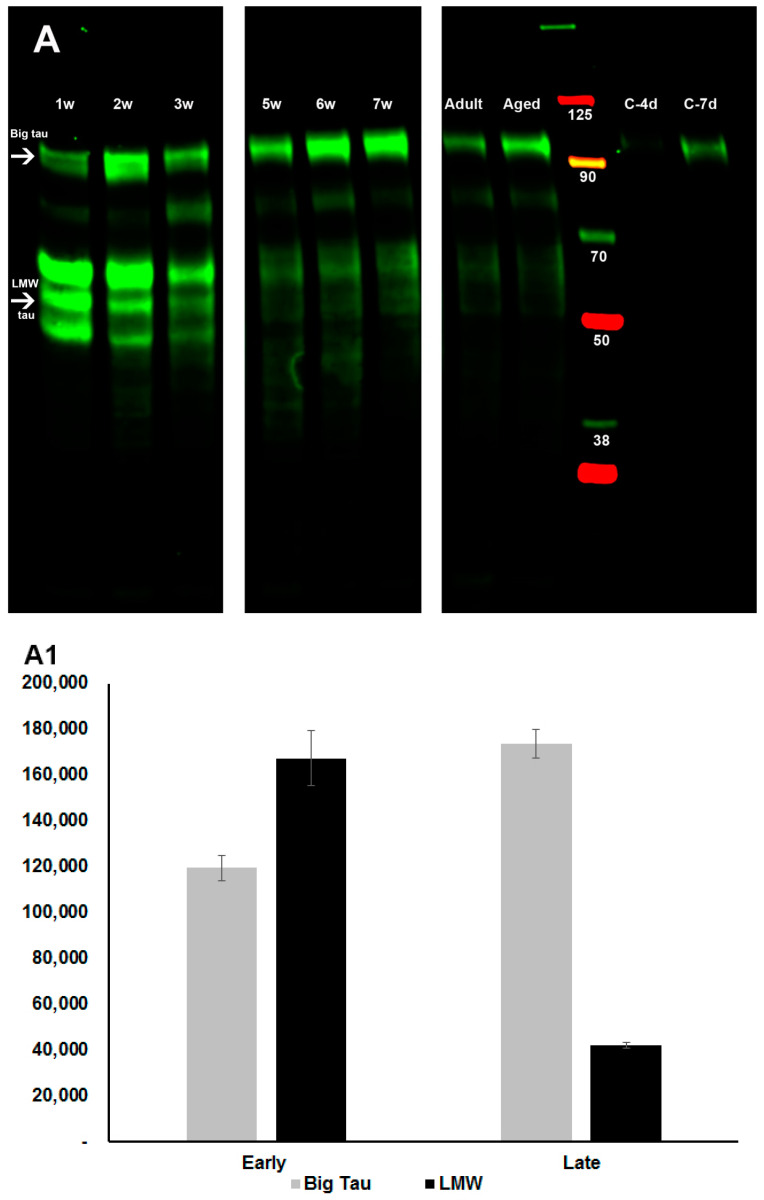
Developmental regulation of Big tau expression in SCG. Western blot analysis of SCG tissue prepared from different ages including 1–3, and 5–7 postnatal weeks, adult at 4 months, and aged animals at 12 months, as well as dissociated SCG cultures from week-5 postnatally cultured for 4 and 7 days. Samples were stained with the 3′tau antibodies, which recognize all tau isoforms as described in Methods. The blot (**A**) shows that the LMW (45–60 kDa) tau isoforms are present at postnatal weeks 1–3 of SCG together with Big tau at 110 kDa. The levels of the LMW tau isoforms gradually decreased until Big tau became the dominant isoform at 5–7 postnatal weeks as well as in adult and aged animals. (**A1**) shows the quantitative analysis of band densities, depicting the relative levels of Big tau and LMW tau at early postnatal stages (average of weeks 1–3) and late postnatal stages (average of weeks 5–7) using raw data corrected relative to actin shown in (**B**) (e.g., the 2 w lane was overloaded and the C-4d underloaded). A parallel set of Western blots with identical samples were stained with Big tau antibodies (**B**) confirming that the expression of Big tau in SCG increased during early postnatal stages and remained the dominant isoform in SCG in adult and aged animals, as well as cultured dissociated SCG neurons. C-4d, C-7d: dissociated SCG neurons from week-5 that were cultured for 4 and 7 days, respectively. The blots included a pre-stained protein ladder and were probed for expression of actin (42 kDa). The levels of actin were used to normalize loading variations in assessment of changes in the level of expression of the different tau isoforms (shown in (**B**) as red bands and used for both (**A**,**B**)).

**Figure 4 cells-12-00226-f004:**
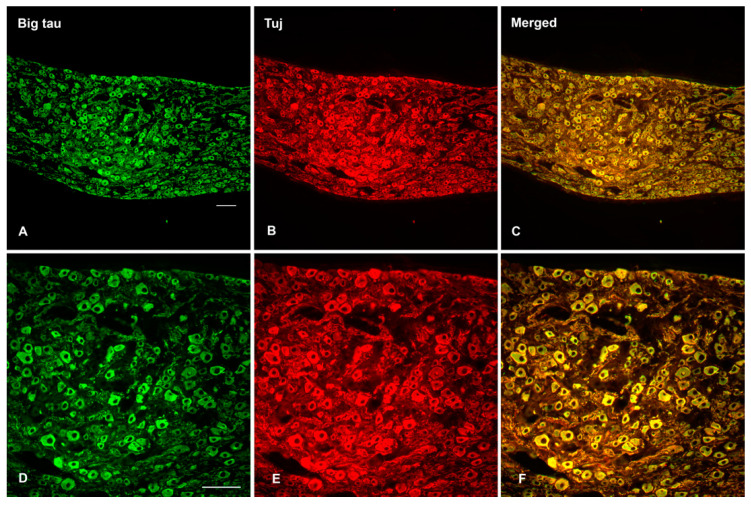
Histological analysis of Big tau expression in adult SCG tissue. Sections from adult SCG were double stained with Big tau and Tuj (neuronal marker) antibodies to analyze the expression and distribution of Big tau expression in SCG. The analysis shows that virtually all SCG neurons expressed Big tau (**A**,**D**) which co-localized with Tuj staining (**B**,**E**), shown in merged image (**C**,**F**). The bottom panel shows high magnificent images derived from the top panel double stained for Big tau and Tuj (**D**–**F**) confirming that all neurons expressed Big tau. Scale bar = 100 µm.

**Figure 5 cells-12-00226-f005:**
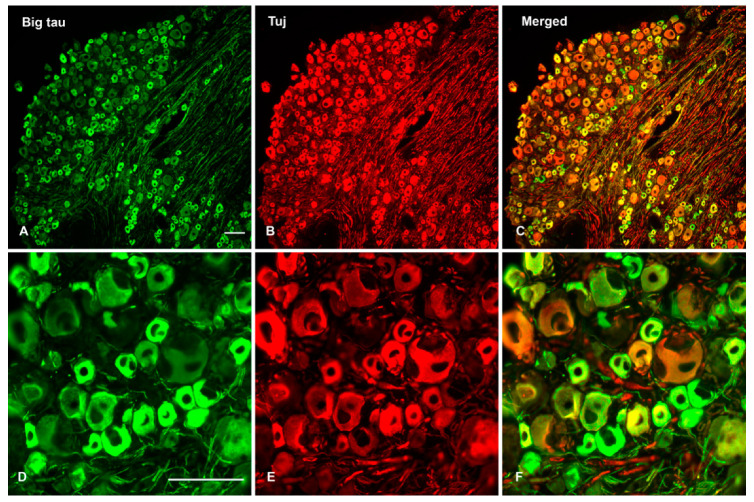
Histological analysis of Big tau expression in adult DRG tissue. Sections from adult DRG were double stained with Big tau and Tuj (neuronal marker) antibodies to analyze the expression and distribution of Big tau expression in DRG. The analysis shows selective staining of DRG neurons with Big tau (**A**,**D**) while Tuj stained all of the neurons (**B**,**E**), shown in merged image (**C**,**F**). The bottom panel shows high magnificent images derived from the top panel double stained for Big tau and Tuj (**D**–**F**) underscoring the selective of Big tau in DRG neurons mostly small and medium-sized neurons as confirmed in Appendix A. Scale bar = 100 µm.

**Figure 6 cells-12-00226-f006:**
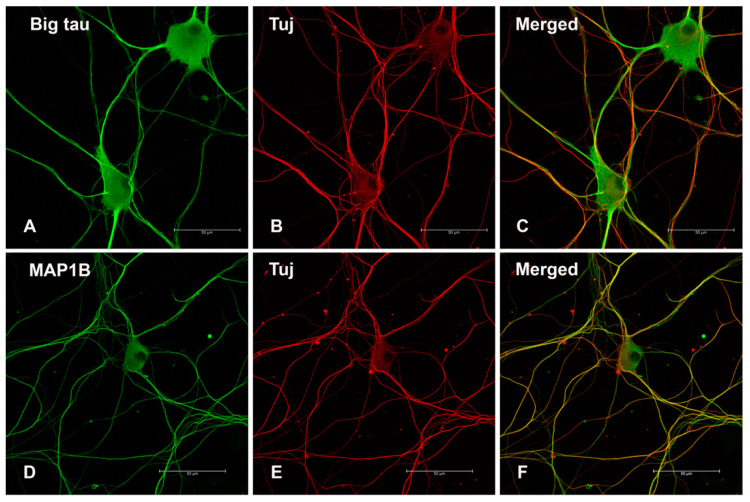
Cultured SCG (P3) stained with Big tau and Tuj. SCGs dissociated from P3 and cultured for 4 days show Big tau expression in both SCG cell body and neurites (**A**), which co-localized with Tuj (**B**,**C**). The SCG cultures showed similar staining with MAP1B (**D**) which also co-localized with Tuj (**E**,**F**). Scale bar = 50 µm.

**Figure 7 cells-12-00226-f007:**
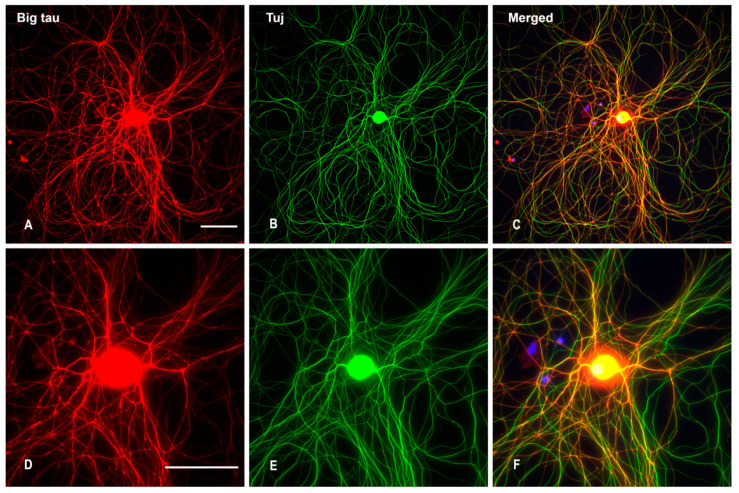
Cultured SCG (P28) stained with Big tau and Tuj. SCGs dissociated from P28 and cultured for 7 days show big tau expression (**A**) in both cell body and neurites, which co-localized with Tuj (**B**,**C**). (**D**–**F**) are higher magnification images corresponding to images (**A**–**C**) above. Scale bar = 100 µm, for all images.

**Figure 8 cells-12-00226-f008:**
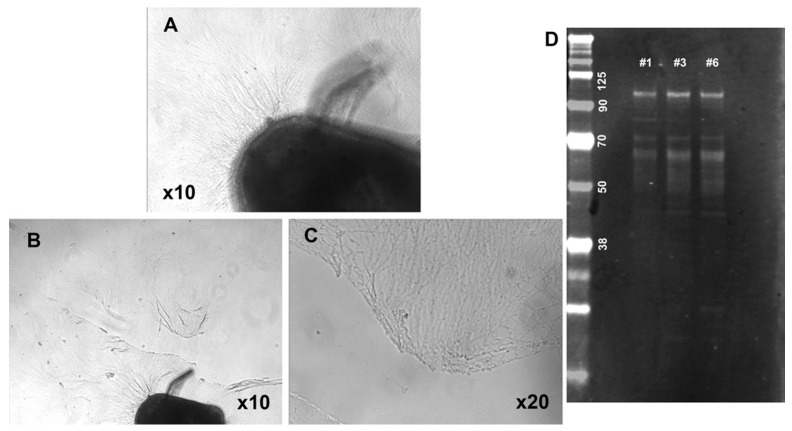
Characterization of SCG explants (P3) cultured for 8 days and lesioned. Phase contrast images show the network of neurites growing around the explant (**A**). They were lesioned at about half of their length (**B**,**C**) and shown at low (**B**) and high (**C**) magnification immediately after the lesion, then allowed to regrow (Figure 9 and Figure 10). Western blot stained with the 3′ tau antibodies show Big tau expression at 110 kDa (**D**) in the P3 explants grown for 8 days, which were lesioned and cultured overnight (lanes #1 and #3) with lane #6 showing the explant without the lesion.

**Figure 9 cells-12-00226-f009:**
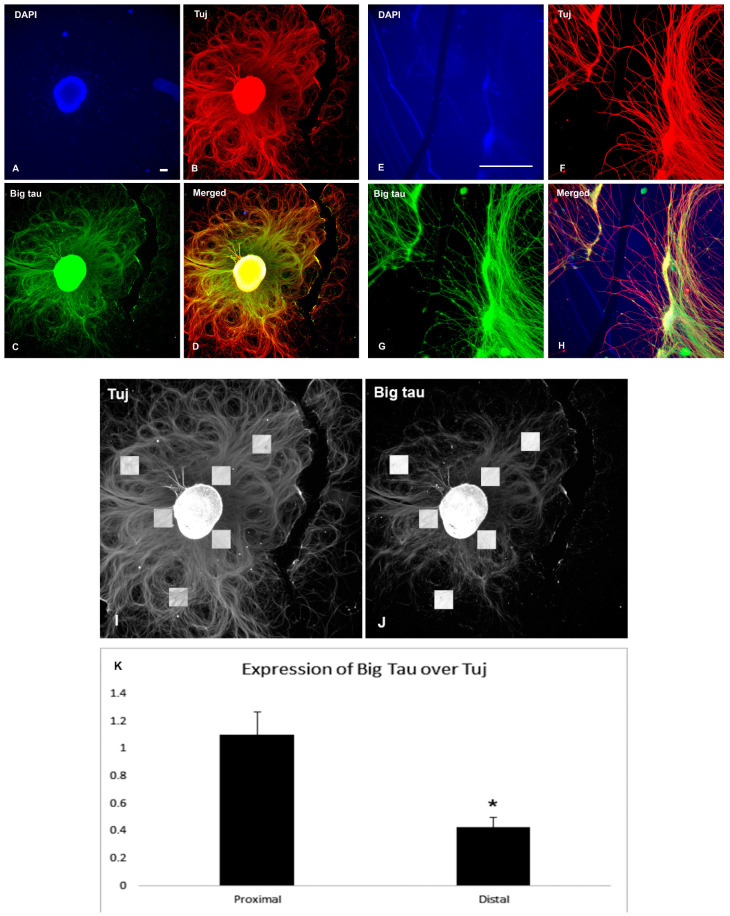
Big tau expression after SCG explant lesion (4 h). The analysis shows 2 sets of P3 SCG explants that were cultured for 8 days, and then grown for 4 h after lesion. Panels (**A**–**D**) show SCG explant lesion at low magnification. Panels (**E**–**H**) show SCG neurites growing near the lesion area at the high magnification. DAPI staining (**A**,**E**), Big tau staining (**C**,**G**), Tuj staining (**B**,**F**), and double staining (**D**,**H**) demonstrates neurite growth and expression of Big tau and Tuj 4 h after the lesion. The analysis of the ratio of Big tau/Tuj 4 h after lesion is based on data presented in (**B**,**C**). Panels (**I**,**J**) indicate the proximal and distal locations (*n* = 3) of the areas included in the average density calculation, respectively, expressed as a ratio of Big Tau over Tuj (**K**). The ratios of Big tau/Tuj densities are significantly lower in the distal explant relative to proximal (* *p* = 0.02). Scale bar = 100 µm.

**Figure 10 cells-12-00226-f010:**
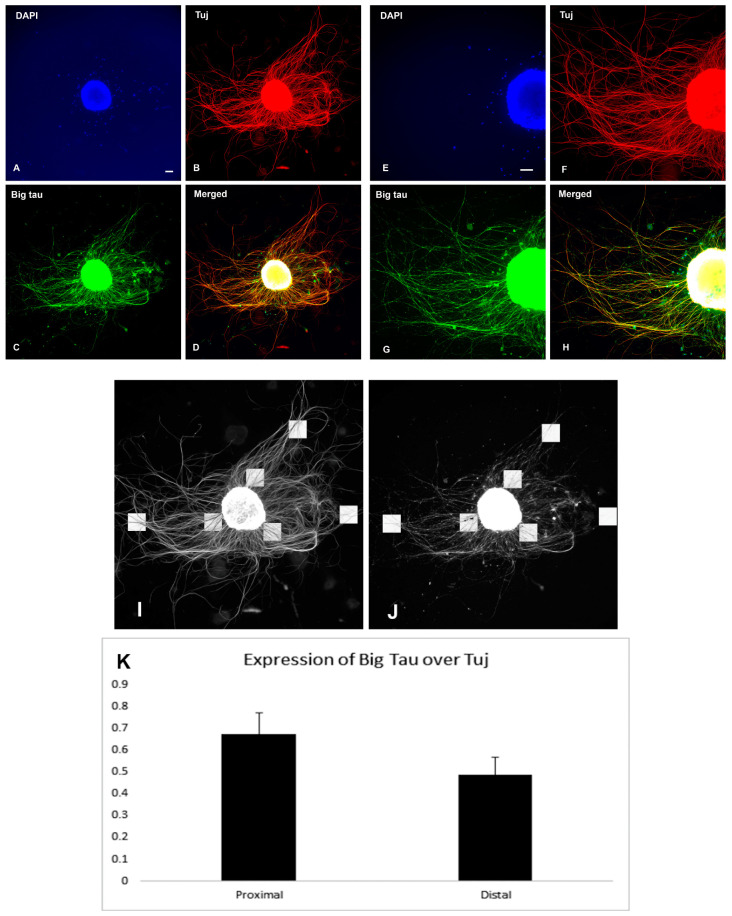
Big tau expression after SCG explant lesion (5 days). P3 SCG explants were cultured for 8 days and then continued to grow for 5 days after lesion. Panels (**A**–**D**) show SCG explant with lesion at low magnification. Panels (**E**–**H**) show SCG neurites growing at the high magnification. The DAPI staining show the explant cell bodies (**A**,**E**), while Tuj (**B**,**F**) and Big tau (**C**,**G**) antibodies also stain the regrowing neurites (merged in (**D**,**H**)). Note that at 5 days post-lesion Big tau is expressed along the entire length of the neurites similar to the staining of Tuj. The analysis of the ratio of Big tau/Tuj 5 days after lesion is based on data on presented in (**B**,**C**). Panels (**I**,**J**) indicate the proximal and distal locations (*n* = 3) of the areas included in average density calculation, respectively, expressed as a ratio of Big Tau over Tuj (**K**). There is no significant difference between the ratios of Big tau/Tuj densities at proximal explant relative to distal (*p* = 0.22). Scale bar = 100 µm.

## Data Availability

All data needed to evaluate the conclusions in the paper are present in the paper.

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
