# Peer review of "Regulation of Tau Expression in Superior Cervical Ganglion (SCG) Neurons In Vivo and In Vitro"

_cells, 2023, doi:10.3390/cells12020226_

Round 1
Reviewer 1 Report
The paper "Regulation of tau expression in superior cervical ganglion 2 (SCG) neurons in vivo and in vitro" is an interesting paper focused on the expression of the Tau isoform named "Big Tau" in SCG and DRG. The authors report that during development neurons change tau expression from the low molecular weight (LMW) isoforms to Big tau. They find Big tau expression in dissociated postnatal SCG neurons suggesting that they are an ideal system to study the function of Big tau in neurons. Finally, they perform preliminary experiments in SCG explants to examine the localization of Big Tau after lesion and regeneration. The field about Big Tau is still open and several aspects are obscure, this paper provides some reliable evidences but I have some concerns about some experiments and the conclusions of the paper.
experimental procedures: tissue preparation: for morphological analysis add the age of the animals and the instrument used to cut samples
line 74: please correct "western blot"
Figure 3: the gels look overexposed. enlarge figure and labels and add a quantification to support your conclusions in results.
Results: 3.3 line 218: "The selective expression is also apparent when the DRG are double stained for Big tau and markers of peptide neurotransmitters (CGRP, IB4) or parvalbumin (data not shown)." Why data not shown? please add them in figure 5 or in supplementary figure, it could be helpful to support the selective expression of Big Tau in DRG. Moreover if possible add a brief comment in Discussion.
Figure 7: panels D, E and F are not described in figure legends. If they are a magnification of panels A,B and C in my opinion you could remove it.
Discussion: from line 356: I think that adding MAP1B staining and Tau colocalization in figures 9-10 could help supporting your hypothesis.
In my opinion the paper is interesting but I think that the authors should improve it by adding an immunostaining of total tau in the regeneration experiment. From western blot experiments Big Tau is the dominant form but tau isoforms are still present. The staining could suggest if normal Tau participates to the regeneration process or it behaves like big tau. Moreover I think that some functional experiments must be provided such as the effect on Big Tau of pathological Tau mutations (for example P301L or others) in SCG. I suggest to add a western blot with big tau mutation to evaluate if it alters the big tau levels in your system during development and a regeneration experiment after lesion. These experiments could provide interesting hints about the mechanisms of big tau. In my opinion these experiments would significantly improve the message and strength of the paper.
Author Response
Reviewer 1
The paper "Regulation of tau expression in superior cervical ganglion 2 (SCG) neurons in vivo and in vitro" is an interesting paper focused on the expression of the Tau isoform named "Big Tau" in SCG and DRG. The authors report that during development neurons change tau expression from the low molecular weight (LMW) isoforms to Big tau. They find Big tau expression in dissociated postnatal SCG neurons suggesting that they are an ideal system to study the function of Big tau in neurons. Finally, they perform preliminary experiments in SCG explants to examine the localization of Big Tau after lesion and regeneration. The field about Big Tau is still open and several aspects are obscure, this paper provides some reliable evidences but I have some concerns about some experiments and the conclusions of the paper.
Thank you for your insightful comments, they have stimulated us for major changes in our manuscript that included revision of several figures, and 2 additional figures in the supplemental section, quantitative analysis of 2 sets of data as well as significant rewriting of all section of the manuscript, which is enclosed with and without tracking.
experimental procedures: tissue preparation: for morphological analysis add the age of the animals and the instrument used to cut samples
We collected SCGs from 1-7 weeks-old rats for the developmental analysis,4 months-old for adult analysis, and >12 months-old for aged animals. These samples have been used for western blots analysis. For immunohistochemical staining we used 4 months-old rats. For dissection, SCG was identified on the underside of a bifurcation of the carotid artery. Under dissecting microscope, the skin at the throat was cut from the chin to the upper chest. The fat pad around the sides of neck was cut away to visualize the carotid artery, three muscles needed to be clipped away by first removing the superior-most muscle (closer to head), then the middle one and finally the muscle lying parallel to the trachea. Using forceps to hold the artery at the point of the bifurcation, SCG was underneath the bifurcation and could be completely removed.
line 74: please correct "western blot"
We changed “Western blot” to “western blot”.
Figure 3: the gels look overexposed. enlarge figure and labels and add a quantification to support your conclusions in results.
We revised Fig. 3 accordingly and added more details with the quantification of the results.
Results: 3.3 line 218: "The selective expression is also apparent when the DRG are double stained for Big tau and markers of peptide neurotransmitters (CGRP, IB4) or parvalbumin (data not shown)." Why data not shown? please add them in figure 5 or in supplementary figure, it could be helpful to support the selective expression of Big Tau in DRG. Moreover if possible add a brief comment in Discussion.
We added the CGRP and IB4 staining in a supplementary figure and additional discussion. We decided to use a supplementary figure because the focus of the paper is on SCG, but we agree with the reviewer that the data will illustrates the selective expression of Big tau in DRG neurons and will underscore the differences between DRG and SCG.
Figure 7: panels D, E and F are not described in figure legends. If they are a magnification of panels A,B and C in my opinion you could remove it.
D, E, and F are higher magnification images related to A-C above with same scale bar. We added that information in the manuscript.
Discussion: from line 356: I think that adding MAP1B staining and Tau colocalization in figures 9-10 could help supporting your hypothesis.
To support our hypothesis, we added quantitative analysis to the distribution of Big tau in proximal and distal region of the SCG explants as described in more details in the manuscript.
In my opinion the paper is interesting but I think that the authors should improve it by adding an immunostaining of total tau in the regeneration experiment. From western blot experiments Big Tau is the dominant form but tau isoforms are still present. The staining could suggest if normal Tau participates to the regeneration process or it behaves like big tau. Moreover I think that some functional experiments must be provided such as the effect on Big Tau of pathological Tau mutations (for example P301L or others) in SCG. I suggest to add a western blot with big tau mutation to evaluate if it alters the big tau levels in your system during development and a regeneration experiment after lesion. These experiments could provide interesting hints about the mechanisms of big tau. In my opinion these experiments would significantly improve the message and strength of the paper.
We emphasize in our discussion that the regeneration experiments in the last part of our paper (Figs 8-10) are preliminary and indeed we plan to continue these experiments in vitro and in vivo. However, we improved the analysis by providing quantitative analysis of Big tau distribution relative to Tuj in proximal and distal areas of the explant. Given the wide scope of our results (identification of Big tau in SCG, analysis of exon structure, in vivo distribution and comparison to DRG, and characterization of dissociated and explant culture), detailed analysis of regeneration will be included in a separate paper.
Reviewer 2 Report
The study performed by Y. Jin and coworkers characterizes the expression levels of tau and particularly the expression from the low molecular weight (LMW) isoforms to Big tau during development. The research field of this study is of particular interest in understanding the role of the Big tau isoform in development. Regarding the experimental data, the claims are not fully supported, so the authors need to provide further evidences, as detailed described below, to support efficiently the statements. The manuscript is well written, in the introduction are included some of the most important papers on the field highlighting the knowledge gap about the developmental changes of tau protein, but the authors should explain with more details those studies emphasizing the main points. Furthermore, in the discussion the interpretation of the results does not reveal their significance for development and tauopathies.
Major points
A general comment regarding all Figures: the authors do not mention et al if they performed three independent replicates for each experiment which is more than necessary so that to reach a conclusion.
Furthermore, from material and methods session is missing the statistical analysis, which gives the impression that the authors do not have the three independent replicates.
Figure 1: it would be easier to follow if the authors show in the image the different isoforms in the corresponding molecular weight, on the right of each image. The same should happen for actin as well.
Any comment why with the big tau specific antibody detects the isoforms with lower molecular weight as well?
Big tau is not detected in the brain extracts. Is this pattern expected?
In addition, a quantification of signal intensities in the expression levels of the different isoforms, or at least in the increase of the big tau needed, as well as normalization to actin, using as control the brain extracts.
Figure 2: The brain extracts is missing as control from the PCR. Furthermore, in panel A.1 lanes 1 and 2, the expected PCR product is 836 and 923bp. How do the authors explain the band around 400bp? The same happen in pane A.2.
In the graphical representation of Big tau please include in the graph the A1, A2 and A3 PCR products.
Is mentioned in the text (Results,3.2): “We found 180 that in the first 2 weeks postnatally we could detect high levels of LMW at 45-60kDa which 181 were reduced at week 3 and remained at low levels at week 5-7 (Fig. 3A). Altogether, the average levels of week 1-3 relative to week 5-7 were reduced by 75%. “ Since a quantification is not provided how do the authors ended up to this conclusion?
Same comments as in Figure 1 about the molecular weights of the different isoforms. A quantification is missing.
The authors must combine images 4 and 5, is easier to follow and compare.
The same for images 6 and 7.
Results 3.5. The authors should provide the biological importance of these set of experiments. In addition, they must include a previous literature of this technique.
The author’s mention: “Explant cultures also allowed us to collect enough tissue for Western blot analysis. Figure 8 demonstrates the expression of Big tau in these cultures (Fig. 8 D). ”
Figure 8D. Actin loading control is missing, also the quantification.
The author’s mention: “Figure 9 shows the result of Big tau expression 4 hours after lesion (Fig. 9 C in both panels left and right) demonstrating regrowth indicated by Tuj (Fig. 9 B in both panels left and right), while the staining with Big tau was present only in the proximal region of the neurites. Five days later the Big tau (Fig. 10 C in both panels left and right) staining overlapped with Tuj (Fig. 10 B in both panels left and right) restoring the original profile of the explant expression .” Figure 9 legend “Note that the distal region of the regenerating neurite express Tuj, a marker of growth, but not Big tau”. Such a statement is not supported from the images. The difference is not obvious. In order to convince about the statement that Big tau was present only in the proximal region of the neurites, a comparison and quantification of the signal intensities with the staining of Tuj in the axons is more than necessary so that to do such a conclusion.
In general, is not well described the purpose of these experiments and the biological relevance of the expected results. In addition, the interpretation of the results in terms of the importance for the development and the correlation with tauopathies.

Author Response
Reviewer 2
The study performed by Y. Jin and coworkers characterizes the expression levels of tau and particularly the expression from the low molecular weight (LMW) isoforms to Big tau during development. The research field of this study is of particular interest in understanding the role of the Big tau isoform in development. Regarding the experimental data, the claims are not fully supported, so the authors need to provide further evidences, as detailed described below, to support efficiently the statements. The manuscript is well written, in the introduction are included some of the most important papers on the field highlighting the knowledge gap about the developmental changes of tau protein, but the authors should explain with more details those studies emphasizing the main points. Furthermore, in the discussion the interpretation of the results does not reveal their significance for development and tauopathies.
Thank you for your comments, they have stimulated us for major changes in our manuscript that included revision of several figures, and 2 additional figures in the supplemental section, quantitative analysis of 2 data sets as well as significant rewriting of all section of the manuscript, which is enclosed with and without tracking.
We appreciate that the reviewer recognized the gap in knowledge about Big tau, which is the tau form with a major structural modification relative to “conventional tau”, is expressed in the PNS and selective neurons in the CNS, is developmentally regulated and may have important implications in tauopathies. According to reviewer suggestions we improved the interpretation of the results and the discussion citing new evidence from a paper that we published last week (Fischer I (2022) Evolutionary perspective of Big tau structure: 4a exon variants of MAPT. Front. Mol. Neurosci. 15:1019999. doi: 10.3389/fnmol.2022.1019999).
Major points
A general comment regarding all Figures: the authors do not mention et al if they performed three independent replicates for each experiment which is more than necessary so that to reach a conclusion. Furthermore, from material and methods session is missing the statistical analysis, which gives the impression that the authors do not have the three independent replicates.
We very much appreciate the importance of experimental rigor. As shown below, we now include quantitative analysis for Fig 3 (expression of Big tau during SCG development) and Fig 9 and 10 (distribution of Big tau in SCG explants). All of the experiments have been performed multiple times for both the SCG and DRG immunostaing of tissue and cultured cells. In fact, we added another supplementary figure that confirms the selective expression of Big tau in subpopulations of DRG. We made a note on reproducibility of multiple analyses in the Methods section.
As for Fig. 3, we emphasized that the changes in Big tau expression during development are designed to show the trend of decrease in LMW tau expression and increased Big tau. Following your comments, we generated quantitative analysis of the data using an average of 3 samples for each data point (see revised Fig. 3 with graph). The analysis showed that “the average levels of week 1-3 relative to week 5-7 were reduced by 75%”. This was done by measuring the density of either all of the LMW (45-65kDa) bands or the Big tau band (110KDa) with average of 3 samples for early postnatal (week 1,2,3) vs late postnatal (week 5,6,7). We specifically did not show a detailed a time dependent curve of week by week. Furthermore, these data are consistent with previous publications (Black et al., 1996) and are designed to show the general profile of Big tau expression during postnatal time in which many studies prepare cultured SCG neurons without taking into account the presence of Big tau.
Figure 1: it would be easier to follow if the authors show in the image the different isoforms in the corresponding molecular weight, on the right of each image. The same should happen for actin as well.
We added arrows for the corresponding MW.
Any comment why with the big tau specific antibody detects the isoforms with lower molecular weight as well?
These are minor degradation products. Note that we used polyclonal antibodies prepared against the full 4a exon in a bacterial expression vector (Boyne et al., 1995). Consequently, they do not have reactivity to the rest of the tau protein (e.g., LMW form). We added this information in methods.
Big tau is not detected in the brain extracts. Is this pattern expected?
As we published before (Boyne et all 1995), Big tau is expressed only in selective (and small) areas of the central nervous system (brain and spinal cord) that project to the periphery such as cranial nerve motor nuclei, selective neurons in the cerebellum and retinal ganglion cell and lower motor neurons. So only if we enrich for these regions or overload the gels, we see a faint band of Big tau (we have the data). Again, we added this information to the paper.
In addition, a quantification of signal intensities in the expression levels of the different isoforms, or at least in the increase of the big tau needed, as well as normalization to actin, using as control the brain extracts.
Figure 2: The brain extracts is missing as control from the PCR. Furthermore, in panel A.1 lanes 1 and 2, the expected PCR product is 836 and 923bp. How do the authors explain the band around 400bp? The same happen in pane A.2.
In the graphical representation of Big tau please include in the graph the A1, A2 and A3 PCR products.
Corrected.
Is mentioned in the text (Results,3.2): “We found 180 that in the first 2 weeks postnatally we could detect high levels of LMW at 45-60kDa which 181 were reduced at week 3 and remained at low levels at week 5-7 (Fig. 3A). Altogether, the average levels of week 1-3 relative to week 5-7 were reduced by 75%. “ Since a quantification is not provided how do the authors ended up to this conclusion?
See our response above.
Same comments as in Figure 1 about the molecular weights of the different isoforms. A quantification is missing.
The authors must combine images 4 and 5, is easier to follow and compare.
The same for images 6 and 7.
We tried to combine the figures and found that they became complicated with too many panels (12 for Fig. 4+5 and 18 for Fig. 6+7) and therefore prefer to keep them separate.
Results 3.5. The authors should provide the biological importance of these set of experiments. In addition, they must include a previous literature of this technique.
We did discuss the importance of the SCG culture data (Fig. 6, 7) and cited the studies that used SCG cultures. “Besides the general characterization of tau expression in SCG we expect that this work will also align the results and interpretation of cell biological studies using SCG cultures with the expression of the specific isoforms of tau. For example, a standard protocol of SCG cultures is using neurons derived from early postnatal time points when they express both LMW tau and Big tau (Baas et al., 1994;Black et al., 1996;Yang et al., 2007;Kawataki et al., 2008;Jean et al., 2012;Majdazari et al., 2013;Pellegrino and Habecker, 2013;Wehner et al., 2016). In these and many other studies few have considered the unusual properties of SCG cultures where microtubules are associated with the unique isoform of Big tau even when studying axonal properties closely related to cytoskeleton structure and function”. Nevertheless, in response to both Reviewer 1 and 2 we improved our discussion relative to physiological function during development and the pathology represented in tauopathies.
The author’s mention: “Explant cultures also allowed us to collect enough tissue for Western blot analysis. Figure 8 demonstrates the expression of Big tau in these cultures (Fig. 8 D). ”
Figure 8D. Actin loading control is missing, also the quantification.
The analysis in Fig. 8D was designed to confirm that our cultures expressed Big tau, that is all. We did not make quantitative statements on the relative amounts of Big tau.
The author’s mention: “Figure 9 shows the result of Big tau expression 4 hours after lesion (Fig. 9 C in both panels left and right) demonstrating regrowth indicated by Tuj (Fig. 9 B in both panels left and right), while the staining with Big tau was present only in the proximal region of the neurites. Five days later the Big tau (Fig. 10 C in both panels left and right) staining overlapped with Tuj (Fig. 10 B in both panels left and right) restoring the original profile of the explant expression .” Figure 9 legend “Note that the distal region of the regenerating neurite express Tuj, a marker of growth, but not Big tau”. Such a statement is not supported from the images. The difference is not obvious. In order to convince about the statement that Big tau was present only in the proximal region of the neurites, a comparison and quantification of the signal intensities with the staining of Tuj in the axons is more than necessary so that to do such a conclusion.
Thank you for drawing our attention to the shortcoming of the analysis and interpretation of Figure 9 and 10. Following your suggestion we did “a comparison and quantification of the signal intensities with the staining of Tuj in the axons” and found that indeed the ratio of Big tau/Tuj 4 hours after lesion was significantly lower in distal relative to proximal regions at about 39% levels. In contrast, although the ratio of Big tau/Tuj 5 days after lesion was still low in distal relative to proximal regions at about 72% levels, the difference it was not significantly different. However, we emphasized in our discussion that the regeneration experiments in the last part of our paper (Figs 8-10) are preliminary and indeed we plan to continue these experiments in vitro and in vivo. However, given the wide scope of our results (identification of Big tau in SCG, analysis of exon structure, in vivo distribution and comparison to DRG, and characterization of dissociated and explant culture), detailed analysis of regeneration will be included in a separate paper.
In general, is not well described the purpose of these experiments and the biological relevance of the expected results. In addition, the interpretation of the results in terms of the importance for the development and the correlation with tauopathies.
In repones to the reviewers we added more discussion about the importance for the development and the correlation with tauopathies.
Round 2
Reviewer 1 Report
The authors emproved the manuscript in several aspects. I appreciate the changes. I have minor concerns:
- In general, in all the graphs, add the number of replicates in figure legend.
- In figure 3 the graph is too small and overlying the blot image (I don't know if it is a formatting error of the file). I suggest to enlarge the graph or labels and cut the low molecular weights (where no signal is detected) in blots.
-I suggest to include figure S2 and S3 in the main text.
Author Response
1) We added the replication data in Methods and Legend.
2) We enlarged the graph in Figure 3
3) We moved the quantitative graphs from S2 ans S3 to main text.
Thank you very much for your comments which have helped us to improve our manuscript.
Reviewer 2 Report
The paper is accepted in its present form.
Author Response
Thank you very much for helpful suggestions which helped us to imporve our manuscript.